# IoT System Based on Artificial Intelligence for Hot Spot Detection in Photovoltaic Modules for a Wide Range of Irradiances

**DOI:** 10.3390/s23156749

**Published:** 2023-07-28

**Authors:** Leonardo Cardinale-Villalobos, Efren Jimenez-Delgado, Yariel García-Ramírez, Luis Araya-Solano, Luis Antonio Solís-García, Abel Méndez-Porras, Jorge Alfaro-Velasco

**Affiliations:** 1School of Electronic Engineering, Costa Rica Institute of Technology, Cartago 159-7050, Costa Rica; 2School of Computer Engineering, Costa Rica Institute of Technology, Cartago 159-7050, Costa Rica; 3School of Physics, Costa Rica Institute of Technology, Cartago 159-7050, Costa Rica

**Keywords:** IoT System, infrared thermography, deep learning, machine learning, random forest, Mobilenet, Resnet50, photovoltaic installation

## Abstract

Infrared thermography (IRT) is a technique used to diagnose Photovoltaic (PV) installations to detect sub-optimal conditions. The increase of PV installations in smart cities has generated the search for technology that improves the use of IRT, which requires irradiance conditions to be greater than 700 W/$m^{2}$, making it impossible to use at times when irradiance goes under that value. This project presents an IoT platform working on artificial intelligence (AI) which automatically detects hot spots in PV modules by analyzing the temperature differentials between modules exposed to irradiances greater than 300 W/$m^{2}$. For this purpose, two AI (Deep learning and machine learning) were trained and tested in a real PV installation where hot spots were induced. The system was able to detect hot spots with a sensitivity of 0.995 and an accuracy of 0.923 under dirty, short-circuited, and partially shaded conditions. This project differs from others because it proposes an alternative to facilitate the implementation of diagnostics with IRT and evaluates the real temperatures of PV modules, which represents a potential economic saving for PV installation managers and inspectors.

## 1. Introduction

Solar photovoltaic (PV) energy and distributed generation are being vastly adopted by smart cities [1]. This energy has considerable advantages; however, it requires correct management to achieve its maximum exploitation and efficiency. The exposure of solar modules to shadows or soiling, or internal damage to the modules, generates a reduction in system performance [2]. These sub-optimal operating conditions must be addressed quickly and effectively to avoid permanent damage to the system, loss of project profitability, or even the possibility of fire being generated [3].

To detect sub-optimal conditions, we can make use of infrared thermography (IRT). This technique allows us to analyze the temperature distribution of the PV arrays to detect regions that are at higher temperatures (hot spots), indicating that there is a loss of power (sub-optimal condition) [4]. Hot spot detection has traditionally been performed manually. In PV modules, they are identified through the presence of a temperature gradient of at least 10 °C. However, for the same sub-optimal condition, the temperature gradient becomes greater as the irradiance increases [5]. The author in [6] proposes a method that should be performed with an irradiance of at least 600 W/$m^{2}$ and [7] indicates that at least 700 W/$m^{2}$ is required. The existence of multiple fault detection criteria makes the diagnosis more complex and generates a loss of objectivity.

Costa Rica is in a tropical latitude with a very variable climate due to its mountainous topography, presenting cloudy or rainy conditions even in the dry season. This climate variation, to which many other countries in the world are also exposed, makes it difficult to ensure conditions of at least 600 W/$m^{2}$ at the time of an IRT inspection. Thus, there is a need to define new hot spot detection criteria under lower irradiances [8,9]. Since the temperature gradient of a hot spot decreases as the irradiance decreases [5], this project proposes a system capable of associating hot spot temperature gradients of less than 10 °C under irradiance conditions lower than 700 W/$m^{2}$, by generating a model based on AI trained with hot spot conditions for a wide range of irradiances. Training this model requires the processing of a large amount of IRT data, including information under sub-optimal conditions and multiple irradiance conditions. Unfortunately, there are few freely available or freely accessible image data with the aforementioned information [10]; therefore, as a first step, it is necessary to collect a large amount of IRT data from PV modules under sub-optimal conditions.

To generate a high volume of thermographic information, this research group developed the project described in [11]. In that project, an IoT-based platform was developed to capture thermal images of PV modules automatically and remotely, monitoring the irradiance and environmental information required for correct thermal image processing. In the research, the correct operation of the system was validated, and a detailed extraction of the PV array temperature information performed, with the aim of automatically detecting sufficient IRT data hot spots even under irradiances lower than 700 W/$m^{2}$.

Accurate detection of solar modules in IRT images has numerous practical applications [12]. For example, it can be used in the monitoring and maintenance of large-scale solar installations, enabling early detection of faulty or mispositioned modules. IRT images can provide valuable information about their status and performance. However, manual processing of these images is a slow and error-prone process.

This is where deep learning comes into play, a branch of AI that has revolutionized computer vision. In this article, we explore the use of deep learning and pre-trained models, specifically, ResNet50 [13] and MobileNet [14], for the detection of solar modules in IRT images. These pre-trained models are convolutional neural networks [15] that have been trained on massive and diverse datasets, enabling them to learn general and complex features that can be applied to specific tasks, such as object detection.

The ResNet50 [13] model is known for its depth and ability to capture high-quality features, making it a promising choice for the detection of solar modules in IRT images. On the other hand, MobileNet stands out for its computational efficiency and ability to run on resource-constrained devices, making it suitable for real-time implementations. The process of detecting solar modules using deep learning consists of several stages [16].

This paper presents the continuation of the project described in [11]. Through a case study, an algorithm based on AI is proposed for the automatic detection of PV modules and hot spots for irradiance conditions higher than 300 W/$m^{2}$. The system was validated under the following three types of sub-optimal conditions: partial shading, short-circuit in a PV module, and soiling. The results demonstrate the possibility of detecting hot spots even under irradiances lower than 700 W/$m^{2}$, which could significantly reduce maintenance costs of IRT and speed up the detection of sub-optimal conditions. In addition, the information being collected can be shared with other researchers so that they can investigate this topic, a situation that is currently not possible due to the unavailability of open data.

The article is structured as follows: (a) a section on related work with a similar approach is presented; (b) a section on materials and methods where the technical characteristics of the equipment, the system used, and the methodology are described; (c) a section on results and a discussion presenting the experimental results with their interpretation; and, finally, (d) the conclusions of the work.

## 2. Related Work

Previous research efforts have focused on different approaches for hot spot detection in PV modules. Some studies have utilized IRT techniques to capture temperature variations across the module surface, while others have explored the use of machine learning algorithms for classification and anomaly detection. However, existing methods often face limitations in terms of accuracy, scalability, and adaptability to varying irradiances. This motivates the need for an IoT system that can provide real-time monitoring and detection of hot spots in PV modules, capable of handling a wide range of irradiances commonly encountered in outdoor environments.

### 2.1. IoT and Thermography

The development of correct monitoring for the operation and maintenance of a PV system is implemented through monitoring technology and continuous analysis of the system [17]. Among the variety of techniques for PV module diagnostics, the ability of the IRT to detect hot spots is noted [18,19]; therefore, extensive research has been carried out on the correct implementation of this method. For instance, in ref. [20], the performance of PV arrays in the presence of faults was analyzed using a PV analyzer, an electrical power meter, and an IRT camera. Moreover, with the innovation of IoT technology, ref. [21] demonstrated the effectiveness of this technology for PV array diagnosis; in this particular case, using the K-means clustering algorithm. IRT has also been implemented for the development of orthomosaics, demonstrating its ability to detect hot spots, soiling, and cracks semi-automatically [22]. To generate quantitative indicators that characterize IRT, ref. [23] established a metric that takes into account the capabilities of the technique. In addition, ref. [24] makes a comparison between IRT and other fault detection techniques.

For the detection of hot spots with IRT, studies have been done to evaluate the temperature gradient required for specific irradiance conditions. One such example, [25], validated that, for certain irradiance conditions, a temperature gradient greater than, or equal to 10 °C, is required. Related to this, although IRT has improved over time, it its effectiveness in detecting hot spots should be improved, and detection criteria should be generated for facilities where irradiance may decrease unexpectedly [8].

### 2.2. Deep Learning

Deep Learning, a subfield of Machine Learning, has revolutionized the field of computer vision in recent years. In [26], it was proven to be highly effective in object detection and classification in images, making it a promising tool for addressing solar panel detection in IRT images. This study proposes an approach based on deep convolutional neural networks for the detection and localization of solar modules in aerial images. The method achieves high accuracy in detecting solar modules in different scenarios. The research document Deep Learning-Based Detection of PV modules in aerial images presents a system for detecting solar modules using a deep convolutional neural network. The approach achieves accurate and efficient detection of solar modules in high-resolution aerial images [27].

Fault detection in photovoltaic modules with Deep Learning was proposed by [28]. The study used a drone with an IRT camera to locate faults in solar modules. For detection, it used an enhanced version of You Only Look Once (YOLOv3-tiny), storing the information on a remote server via Long-Term Evolution (LTE) communication. The study found that the YOLOv3-tiny model performed well.

Semantic Segmentation of Solar PV modules and Wind Turbines in Satellite Images Using U-Net, presented by [29], focuses on automating the surveying process of renewable energy resources to eliminate manual effort. Specifically, it aims to gather information on Solar PV modules and Wind Turbines at the building or neighborhood level to facilitate the analysis of their deployment. By leveraging a Deep Learning model, it becomes possible to approximate the carbon footprint and payback period for both existing and proposed locations of these renewable energy installations. The Dataset used in this work was acquired from Google Maps, and specifically from its satellite view. Deep convolutional neural network for automatic detection of damaged PV cells [30] introduces a novel approach utilizing Deep Convolutional Neural Networks (DCNNs) to estimate the degradation of PV cells. While numerous studies have focused on image classification, this research stands out as the first to leverage data obtained from a drone equipped with an IRT sensor. The authors present experiments conducted on the PV images Dataset, a collected dataset, to highlight the degradation issue and provide a comprehensive evaluation of the proposed method. Results, in terms of precision, recall, and F1-score, demonstrate the effectiveness and suitability of the approach.

### 2.3. Machine Learning

A commonly used technique is supervised learning, where a model is trained using labeled datasets containing information about the presence or absence of solar modules and their condition. PV defect classification through IRT imaging using a Binary classification learning approach is presented in [31]. This study explores a deep learning and feature-based approach to detect and classify defective PV modules using IRT images in a South African context. The performances of the VGG-16 and MobileNet models are evaluated and found to be effective in accurately classifying defects. To identify the defective PV modules, the scale invariant feature transform (SIFT) descriptor is combined with a random forest classifier. The implementation of this approach shows potential for reducing costs associated with defect classification compared to current methods.

In [32], a binary classification of defective solar PV modules using IRT project is presented; the focus was on addressing the crucial aspects of monitoring and maintenance to ensure the reliability of PV modules within solar PV-based power systems. They proposed a machine learning-based approach that leverages texture features extracted through IRT assessment to categorize operational solar modules into two distinct categories: defective and non-defective modules. Fault Detection and Classification in PV Arrays Using Machine Learning Algorithms in the Presence of Noisy Data is presented in [32]. In this study, the objective was to develop a robust method for fault detection in PV arrays by incorporating machine learning techniques while handling noisy data. The presence of noise in the data poses a challenge to the training process of machine learning algorithms. Furthermore, the data collected from PV arrays may contain noise due to human errors or equipment malfunctions. Therefore, this research focused on devising a methodology to effectively detect faults in PV arrays by mitigating the impact of noisy data.

In [33], a machine learning scheme for anomaly detection in solar power plants is presented. The primary focus is on leveraging recent advancements in machine learning to tackle the challenging task of anomaly detection in PV systems. With the rapid growth of the solar energy industry and the increasing demand for reliable and efficient PV systems, accurately detecting and monitoring anomalies has become a crucial requirement.

### 2.4. Innovation of This Work

In previous projects on hot spot detection in PV installations using AI for IRT, irradiance has not been taken into account [28,31,32], or it the irradiance conditions have been indicated as at a maximum [30], making it impossible to include this variable in the IRT analysis, which is known to be directly related to the hot spot temperature [5]. On the other hand, the above-mentioned projects have not taken into account the specific record for each thermogram, the relative humidity and ambient temperature conditions [28,30,31,32], thereby impeding the inclusion of this information within the AI model, even though these variables have a relationship with the temperature estimated from IRT [34]. In previous studies [28,30,31,32], the detection of hot spots was conducted by considering as the input variable in training AI models the thermograms of the PV installation, but not the analysis of temperature deltas between modules, which is the criterion taken as reference in the IRT analysis [7]. Finally, in the studies mentioned the analyzed thermograms included elements of the surroundings (roads, soil, and vegetation), introducing elements with thermal patterns that could generate false detection of hot spots.

In this project, real-time information on irradiance, ambient temperature, and relative humidity is introduced for each thermogram to achieve a more accurate model that allows the detection of hot spots even under low irradiance conditions. In addition, the AI model takes a validated quantitative criterion for hot spot detection, using, as a reference, the temperature gradients between modules. Finally, so that the analysis performed omits the consideration of unwanted information (elements that do not correspond to hot spots), the detection of the PV modules is introduced as a first step so that the subsequent IRT analysis focuses specifically on the PV modules.

## 3. Materials and Methods

This article presents, in detail, the methodology applied to achieve a new system to detect hot spots for irradiances greater than 300 W/$m^{2}$ using AI. The materials and methods considered in the research are detailed below.

First, thermograms of the PV array are captured using the IRT camera to obtain the thermal information of all PV modules. These images are then processed to identify and isolate each PV module. This step involves thresholding techniques and segmentation algorithms to distinguish the pixels corresponding to each PV module. Once each module has been identified, the next step is to determine the temperature associated with each pixel of each PV module. This is achieved by calibrating the IRT camera and correlating the recorded thermal radiation with known temperature values.

The IRT images are collected and prepared, along with corresponding labels indicating the presence or absence of solar modules. Then, a training dataset is used to fine-tune the weights and parameters of the selected pre-trained model. Once the model has been trained, we proceed to the inference stage, where it is applied to new images to detect and locate the solar modules present in them. During this stage, the model leverages its ability to extract relevant features from the images and classify regions of interest as either solar modules or backgrounds using deep learning techniques, such as the ResNet model.

In this project, an approach for detecting hot spots in IRT images and utilizing machine learning techniques, specifically Random Forests, was evaluated. Random forests constitute a versatile and powerful ensemble learning tool that combines several decision trees to make accurate classifications and predictions.

By associating temperatures with each pixel, a quantitative assessment of the module’s temperature distribution can be obtained [35]. Furthermore, the classification of the overall state of the PV module can be performed, based on the temperatures detected. The idea is that the identification of hot spots may indicate potential issues, such as soiling, partial shading, or electrical malfunctions. By employing Random Forests [36], a machine learning model can be trained on labeled datasets, where the number of hot pixels and other relevant features are associated with the corresponding module conditions. This model can then predict the state of a new module based on its hot pixel count.

The utilization of the random forest technique allows for the classification of temperature differences using the corresponding irradiance, and, from that, a determination of whether the condition is optimal or not. Random forests are a supervised machine learning algorithm employed for both classification and regression tasks. To apply random forests in this case, a data set containing information on irradiance, temperature differences, and the classification of whether a hot spot exists or not is necessary. Historical data can be used to train the random forest model.

Once the model is trained, it can be utilized to classify new observations of irradiance and temperature differences, determining whether a hot spot exists or not. The random forest model leverages the learned information during training to make classification decisions based on the input data characteristics. The integration of machine learning with the analysis of hot pixels in IRT images offers a powerful and efficient approach to temperature determination and classification of PV modules. By leveraging the capabilities of Random Forests, accurate predictions and classifications can be made, enabling proactive maintenance and troubleshooting of PV installations.

### 3.1. PV System Analyzed

A PV array of 12 modules in a series was used at the Costa Rica Institute of Technology, with the characteristics shown in Table 1. Figure 1 shows the PV array under study.

### 3.2. Data Acquisition System for Thermographic Analysis

A thermographic data acquisition system developed in the previous stage of the project was used [11]. A FLIR Vue Pro R 336 camera (FLIR LLC, Wilsonville, OR, USA) and a Spektron 210 irradiance sensor (Tritec Energy, Arundel, UK) were used. The temperature of each pixel of the thermograms was obtained with the thermimage package [37], which, considering Plank’s law and Stefan Boltzmann’s relation, requires, as input parameters, the relative humidity and the ambient temperature; therefore, the system includes a sensor for the real-time measurement of these variables. Table 2 and Table 3 show the characteristics of the instrumentation used. The proper functioning of the system was validated in [11].

Using an IRT protocol defined in [38], the camera was placed in a fixed position to maintain the conditions between the taking of IRT images.

### 3.3. System Requirements

These were the requirements for the implemented system:(a)Automatic and synchronous acquisition of all information needed for IRT analysis.(b)Automatic processing of the IRT data. The technique implicitly measures the temperature of each PV module by means of a model, details in [11].(c)The detection of hot spots in the modules should be automatic. For this purpose, the system verifies that there is at least an irradiance of 300 W/m${}^{2}$.(d)Remote information management. The system can store information in data backup, with the possibility of accessing it online. Additionally, all system control is done via a graphical user interface (GUI).

### 3.4. Algorithm for Hot Spot Detection

An algorithm for automatic hot spot detection was defined and implemented in an IoT-based system. Figure 2 shows the developed algorithm and its relationship with the main components of the system.

From Figure 2, it can be identified that the proposed system has the following steps for the detection of hot spots: (1) Data acquisition for IRT, (2) Detection of PV modules by means of deep learning, (3) Reading the temperature of each pixel of the PV modules, (4) Establishment of a healthy reference module, (5) Detection of hot spots by machine learning and (6) Identification of modules with hot spots.

As can be observed in Figure 2, the system inputs are the thermal image, irradiance, relative humidity, and ambient temperature, all acquired from two Raspberry Pi 4. In step 1, the control unit can be configured to take measurements automatically at a given frequency and also manually (IRT analysis is not yet performed at this point). In step 2, the server runs a deep learning process that automatically detects each of the PV modules present in the captured thermal image; this process has, as output, the coordinates of each PV module. In step 3, each of the identified PV modules is analyzed with the Flir script that interprets the temperature of each of the pixels of each image; for this, it makes use of the real-time measurement of humidity and temperature. Additional details of this process are included in [11]. In step 4, a statistical analysis identifies which PV module is the healthiest (the module with the smallest temperature gradient). This step is essential, since the IRT analysis is based on the comparison of objects to be evaluated with a healthy object [4]. In step 5, each PV module is evaluated with respect to the healthy module. In this analysis, the temperature gradient between modules, known as delta temperature (DT) and irradiance, are taken into consideration. DT is calculated with Equation (Equation 1) for each module evaluated with respect to the healthy module.
(1)$$DT=T_{max,UTest}-T_{max,healthy}$$
where $T_{max,UTest}$ is the maximum temperature in the module under test and $T_{max,healthy}$ is the maximum temperature of the healthy module. The detection of hot spots is achieved using a machine learning AI of Ensemble Classifiers as has been used before for IRT [39]. The output corresponds to a data matrix stored in a CSV file that identifies modules with and without hot spots (Step 6).

### 3.5. Fault Detection Criteria

The whole system considers that a failure in a PV module, due to a hot spot, generates a temperature differential of 10 °C or more for a minimum irradiance of 700 W/$m^{2}$ [7]. With this, the proposed algorithm considers that, if, for a given sub-optimal condition, a hot spot is generated at an irradiance of at least 700 W/$m^{2}$, then, for an irradiance lower than 700 W/$m^{2}$ that condition still corresponds to a hot spot, even if the DT is less than 10 °C.

### 3.6. Deep Learning for Solar Module Recognition

The recognition of PV modules using deep learning techniques, and, specifically, the ResNet50 and MobileNet architectures, has proven to be an effective solution for the automated identification of these devices in images. This system utilizes deep learning to analyze and classify images (step 1), enabling accurate and efficient detection of solar modules in various scenarios.

ResNet50 is a deep convolutional neural network (CNN) architecture that has demonstrated exceptional performance in image classification tasks. It consists of 50 layers of learning, allowing it to effectively capture complex and high-level features from images. On the other hand, MobileNet is another CNN architecture known for its computational efficiency and ability to run on mobile devices with limited resources.

The system training process begins with the collection of a labeled dataset of PV modules, consisting of images that contain both solar modules and other features present in the environment. These images are used to train the neural network, where they are fed into the network, and the weights of different layers are adjusted so that the network can learn to recognize and distinguish solar modules.

Once trained, the network can perform inferences on new images to detect and classify PV modules. The system takes an image as input and passes it through the neural network. The initial layers of the network capture low-level features, such as edges and textures, while the later layers capture more abstract and module-specific features.

The ResNet50 and MobileNet architectures are capable of extracting distinctive features of solar modules, such as the structure of PV cells and the frame of the module. These features are used by the system to determine if a solar module is present in the image and to provide an accurate classification of its location and orientation.

The training of the ResNet model was conducted using a dataset of 6800 images of PV modules. To enhance the identification of PV modules’ capability, the technique of knowledge transfer was applied using TensorFlow, a popular machine learning library. Knowledge transfer is an approach that leverages the knowledge previously acquired by a model in a related task to improve performance in a specific task. In this case, a pre-trained model on a general dataset was used to initiate the training of the ResNet model. This pre-trained model benefited from its previous experience in learning relevant image features. The technique of knowledge transfer enabled this knowledge to be transferred to the ResNet model and enhanced its classification capability for sub-optimal and optimal contexts in PV modules. During the training process, the ResNet model adjusted its parameters using the training data. By training the ResNet model on this specific dataset, the goal was to capture the distinctive characteristics of each category and improve the model’s classification capability on new images. The choice of ResNet as the model architecture was significant due to its ability to learn high-level representations of images. ResNet’s architecture includes residual blocks that enable the model to learn deeper and more complex features. This capability was leveraged to enhance the precise identification of PV modules. During training, optimization techniques, such as stochastic gradient descent, were employed to iteratively adjust the weights and biases of the ResNet model. These adjustments were performed over multiple epochs to gradually improve the model’s ability to generalize and accurately classify new images. To evaluate the performance of the model, an independent test dataset that was not used during training was utilized. Evaluation metrics, such as precision and accuracy, were calculated to measure the model’s performance in classifying sub-optimal and optimal conditions accurately.

### 3.7. Machine Learning for Hot Spot Detection

The random forest classification system is a machine learning technique used to classify PV modules with and without hot spots. For this purpose, the algorithm is trained with information from real-time conditions of irradiance and DT. First, a dataset is collected consisting of images of PV modules. These images contain both healthy solar modules and those with possible anomalies or issues. Along with the images, information about irradiance and DT is gathered. Once the dataset is obtained, the random forest classification system is trained. This algorithm utilizes multiple decision trees, where each tree is trained with a random sample from the dataset. During training, each tree performs classification of the PV modules using the information from each pixel present in the image, as well as the irradiance and DT. During the classification stage, the system takes an image of a PV module as input and submits it to each tree in the random forest. Each tree performs an independent classification and produces its prediction. The output of all the trees is then combined to obtain a final prediction.

Employing a random forest classification system brings numerous advantages to this analysis. Firstly, it enables the handling of large datasets efficiently and exhibits robustness against noise or missing data. This resilience allows for reliable and accurate classification results even in real-world scenarios where data imperfections may exist. Moreover, the random forest model excels in capturing complex and non-linear features that are often present in images of PV modules, enhancing the system’s ability to accurately differentiate between normal and defective modules.

By leveraging information regarding irradiance and temperature, the system becomes capable of detecting a range of potential issues within the modules. These may include soiling, partial shading, and short circuits that would affect the overall performance and efficiency of the solar system. Identifying such problems on time empowers maintenance teams to address them promptly, thereby ensuring the optimal functioning of the PV array and maximizing its power generation capacity.

The classification system described herein offers a comprehensive approach to detecting issues in PV modules (Step 5). By considering the analysis of pixel features, temperature differentials, and irradiance and employing a random forest model, it becomes possible to accurately classify modules as no hot spot (0) or hot spot (1). This approach enables efficient maintenance practices, reduces downtime, and maximizes the overall reliability and performance of solar energy systems.

In this project, a total of 664 observations were used to train a random forest model, with 332 observations with hot spots and 332 in normal conditions. The data was split into a 70 percent training set and a 30 percent test set. This meant that 464 observations (70 percent of 664) were used for training, while 200 observations (30 percent of 664) were reserved for testing the model. By employing this configuration, the model was trained on diverse conditions and evaluated on an independent sample, providing a solid foundation for analyzing its performance and generalization in real-world scenarios.

### 3.8. Validation Process

Initially, the correct data acquisition, processing, and visualization of the system were verified. Then, considering the methodology of [8], sub-optimal conditions (soiling, partial shading and module short-circuit) were induced to evaluate the correct detection of hot spots with different irradiance levels and different hot spot causes. Figure 3 shows an example of a test of soiling conditions. Finally, to obtain an indicator of the system’s performance in detecting hot spots, an inferential statistical analysis was performed using ROC analysis [40]; thus, calculating the sensitivity and accuracy of the system.

### 3.9. Limitations of the Project

It should be taken into consideration that this project presents a platform that can detect hot spots from AI; however, its performance is directly related to the training dataset. Due to the defined availability of time for this stage of the project, the dataset used was limited should be increased, and its improvement should be evaluated in future research.

For this first iteration of the system, it was assumed that it would be at a fixed location; therefore, the data set used corresponds to images taken from the same point. The real potential of this system lies in making it work for images taken from variable locations, as a drone would do. Therefore, improving the dataset with these considerations in mind is still pending.

To evaluate the system’s ability to detect sub-optimal conditions, partial shading, soiling, and short circuits were taken into consideration; however, there are other types of faults for which the system’s performance is yet to be evaluated.

Finally, the performance of the system is closely related to the accuracy of the instrumentation; therefore, there is an inherent error in the equipment used.

## 4. Results and Discussion

The main experimental results obtained in the research are discussed and presented below.

### 4.1. Graphical Interface to Operate the System

Figure 4 shows the system interface in operation. It allows the user to visually know the location of each hot spot in the analyzed thermogram and also allows the user to visualize, in detail, all the variables measured and calculations used in the analysis.

### 4.2. Automatic Module Detection

Figure 5 shows the detection of each PV module numbered with an identifier between 1 and 8. As can be seen, from the deep learning algorithm, the system was able to automatically detect the 8 PV modules that were fully observed in the thermogram, identifying them for further temperature analysis. The labeling of 1–8 was not presented to the user. It is included in Figure 5 to exemplify the concept. It can be seen that the rectangle that identifies each module in some cases includes regions behind the array, and there are also cases where there is an overlap between modules, a feature that could be improved in the future.

### 4.3. Automatic Hot Spot Detection

The system response was evaluated based on hot spot detection analysis for induced sub-optimal conditions. Table 4 shows an example of 4 tests performed. For each test, the irradiance, the modules where a hot spot condition was applied, and the modules where the system detected a hot spot were recorded. As shown in Table 4, the system was able to correctly detect hot spot conditions induced by soiling (Figure 6a) and short-circuit conditions (Figure 6b). It is important to emphasize, in this case, that the failures were induced in PV modules that were randomly located and were for irradiance conditions below 700 W/$m^{2}$, demonstrating the capability of the system to effectively detect hot spots due to different conditions under low irradiances.

Table 4 shows the cases described by Figure 7a (soiling) and Figure 7b (no hot spot induced). The system detected a hot spot in module 2 by mistake. In regard to the case presented with Figure 7a., the hot spot existed in module 5, right at the edge adjacent to module 2; this caused the system to detect the hot spot in both modules (5 and 2). This could be corrected in several ways; for example, using the AI of step 2 through segmentation [41]. In regard to the case presented with Figure 7b, the system detected a hot spot by mistake due to a reflection of an object. The object that generated the reflection would have to be relocated or the camera position changed to correct this.

### 4.4. Performance Evaluation

An experiment was developed by inducing sub-optimal conditions to the PV array and analyzing the system response for each test; from the results, the system performance for hot spot detection was determined. For this, 86 tests of sub-optimal conditions and 10 control (non-failure) conditions were applied to the array under test. In this process, each measurement was taken approximately every 15 min. The results of the confusion matrix are shown in Table 5.

A confusion matrix was generated by considering each PV module as an individual test subject. In this analysis, it was considered that, of the 8 modules analyzed, only 7 could be modules with a hot spot, since one was always chosen as a healthy module. Considering this, with the 96 tests performed on the PV array of interest, a total of 672 modules were evaluated.

From the results of Table 5, it was determined that the system had a sensitivity of 0.995 and an accuracy of 0.923. This verified that the system performed correctly to detect hot spots due to different types of sub-optimal conditions at irradiances from 301 W/$m^{2}$ to 1132 W/$m^{2}$ (range of irradiances used in the 96 tests). It can be observed that there was only one case in which the system did not detect the presence of a hot spot, which was associated with the high sensitivity of the system. On the other hand, 18 hot spots were detected in cases where there were none (false positives), which occurred due to a reflection of an object behind the array of interest, and in cases where two hot spots were detected in two adjacent modules that corresponded to a single hot spot. Thus, it was confirmed that a thermographic image acquisition system could be trained to detect hot spots at irradiances greater than 300 W/$m^{2}$, allowing hot spot detection even under varying weather conditions, where unexpected low irradiances may occur. This confers high benefits to managers and inspectors of PV installations, since suspending a scheduled IRT inspection due to a decrease in irradiance represents economic losses. Additionally, early detection of a hot spot, even under low irradiance conditions, can prevent further damage to an installation.

## 5. Conclusions

It was experimentally proven that, from an IoT-based system with AI, it is possible to detect hot spots in PV modules for irradiances above 300 W/$m^{2}$. This contributes to the solution of an existing need to facilitate managers and thermographers of PV installations in speeding up the detection of hot spots that, up to now, require scheduling inspections for irradiances above 700 W/$m^{2}$.

The system achieved a sensitivity of 0.995 and an accuracy of 0.92, which is associated with high system performance; even so, these results can be improved by implementing the variations indicated in Section 4.3.

The results indicate the ability of the system to detect hot spots due to soiling, partial shadows, and short circuits in modules; however, because the system analyzes temperature differentials, it has the potential to detect hot spots due to any other cause.

This research differs from classical IRT processing techniques in that it is not trained to detect temperature patterns but is capable of analyzing the temperature of each pixel of PV modules to detect a hot spot. The results represent the high potential of combining IRT and AI in IoT systems to maximize the requirement for smart cities to make greater exploitation of PV energy.

The performance of the system can be improved by expanding the training data set. In this process, the inclusion of other sub-optimal conditions would allow the system to be validated against a wider range of possible situations that may occur in real installations; however, because the system focuses on the analysis of temperature gradients, the system should even identify sub-optimal conditions for which it has not been trained. All of the aforementioned should be experimentally validated in future work.

This system can be implemented to automatically evaluate the diagnosis of IRT taken with drones; however, experimental tests with thermograms at different altitudes and orientations are still pending in future research.

It is of high interest for PV installation managers to detect not only the presence of a fault, but also the cause of the fault. The system shown could be improved to achieve this by expanding the dataset and training the AI to identify the main types of faults; this remains to be developed in a future project.

The proposed system involves including, in the IRT analysis of PV systems, a detailed monitoring of irradiance, relative humidity, and ambient temperature, to be subsequently processed in a computer system with AI, These aspects are not currently considered in many cases. On the other hand, the system aims to facilitate and speed up the application and analysis of IRT. The inclusion of this type of factor should be analyzed to estimate the benefit for real inspection scenarios and this type of analysis is a pending line of research.

## Figures and Tables

**Figure 1 sensors-23-06749-f001:**
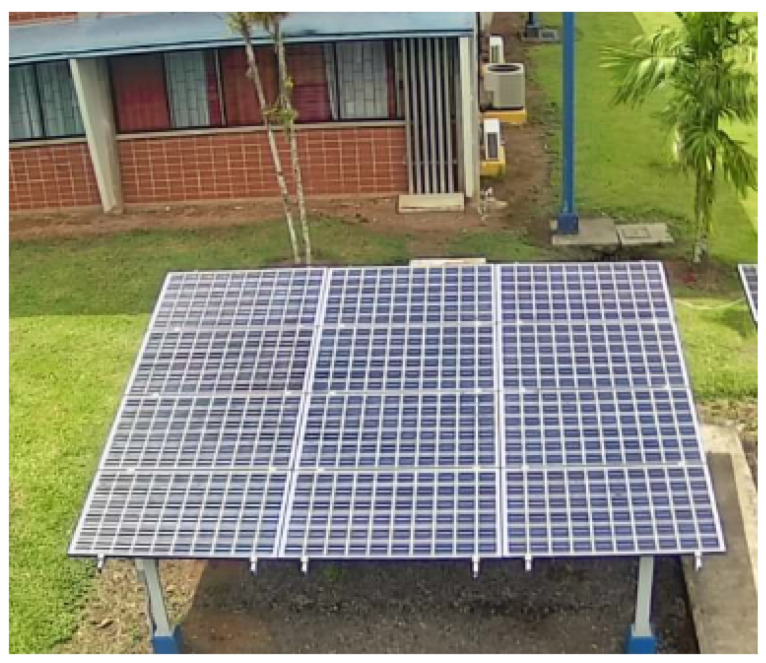
Polycrystalline 3.36 kW PV array used for this study.

**Figure 2 sensors-23-06749-f002:**
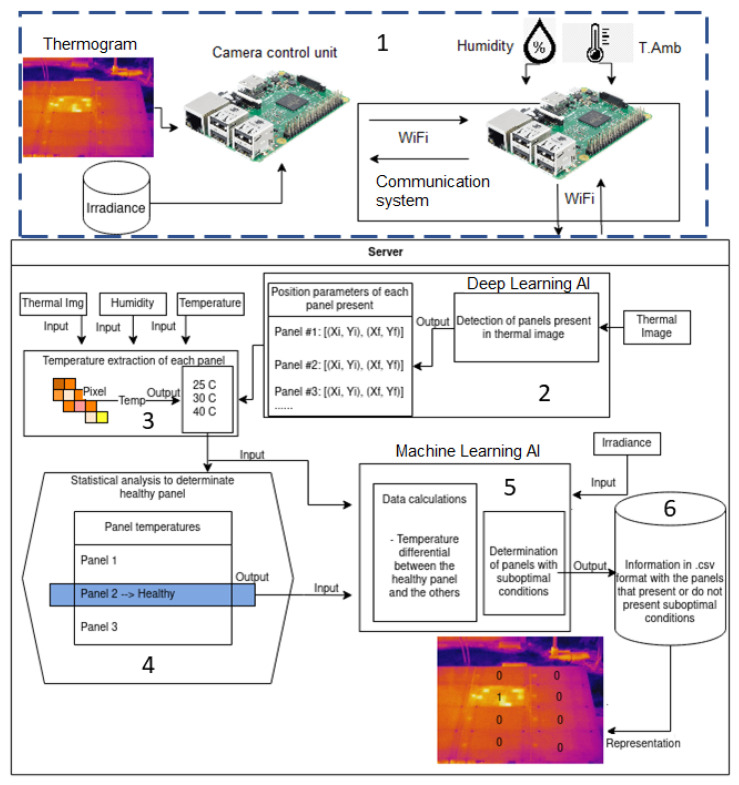
Implemented fault detection algorithm. The flow of information is listed from 1 to 6.

**Figure 3 sensors-23-06749-f003:**
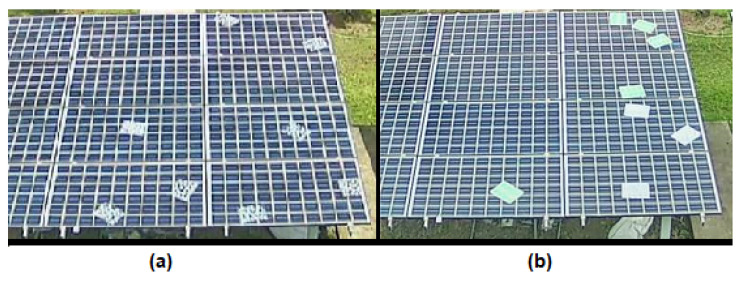
(**a**) Example of application of bird droppings in the PV array under test. (**b**) Example of application of a combination of droppings and leaves in the PV array under test.

**Figure 4 sensors-23-06749-f004:**
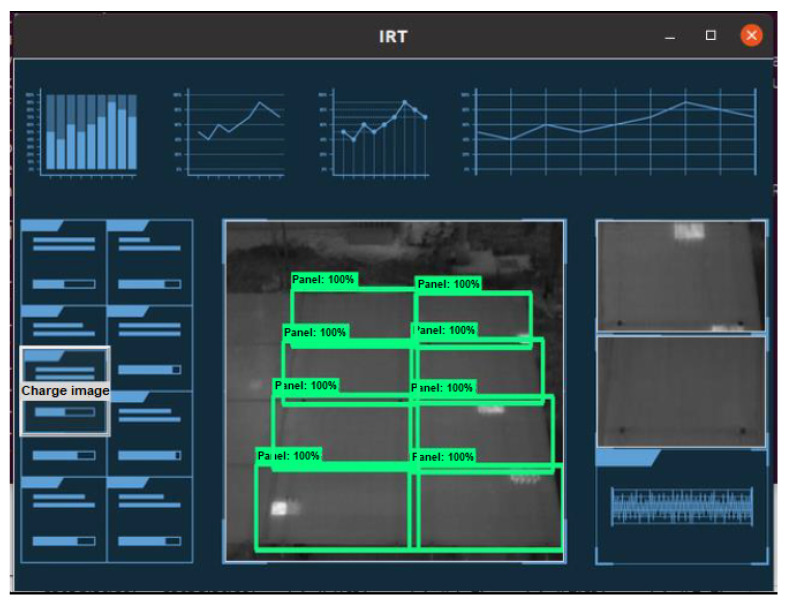
Graphical user interface for presentation of thermographic analysis and hot spot detection.

**Figure 5 sensors-23-06749-f005:**
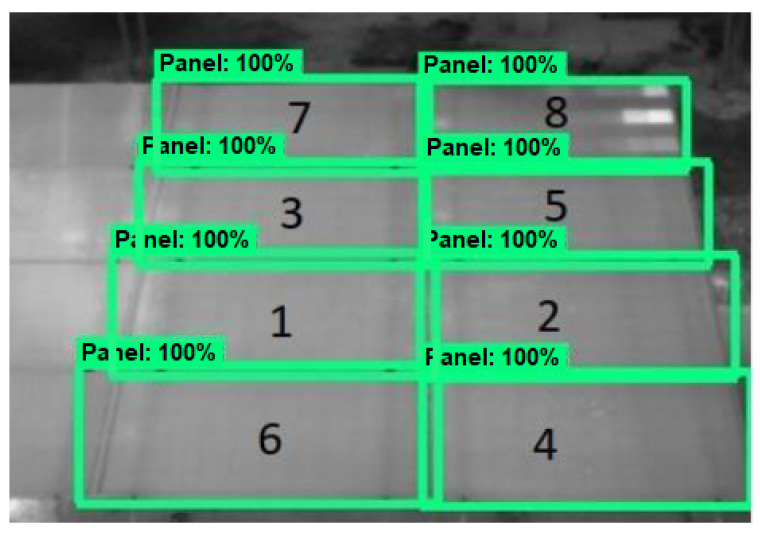
Representation of the automatic identification of PV modules by the system. The numbering was not presented to the user, but was included in the figure to exemplify the process.

**Figure 6 sensors-23-06749-f006:**
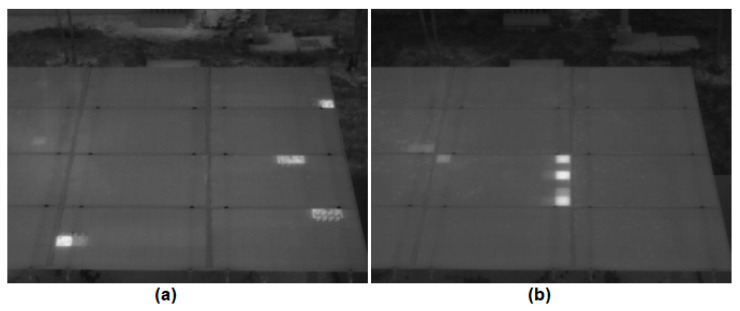
(**a**) Thermogram of array under soiling conditions at an irradiance of 317 W/$m^{2}$. (**b**) Thermogram of array under short-circuit conditions at an irradiance of 430 W/$m^{2}$.

**Figure 7 sensors-23-06749-f007:**
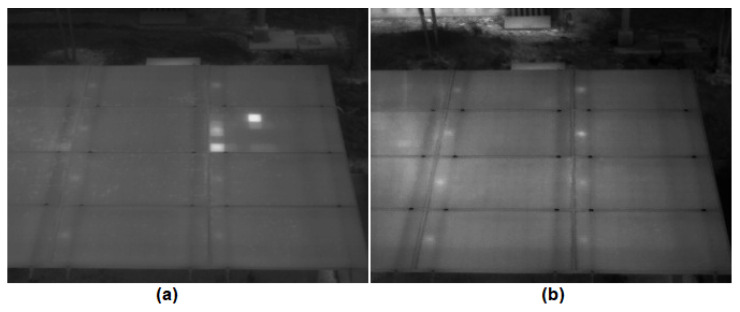
(**a**) Thermogram of array under soiling condition at irradiance of 456 W/$m^{2}$. (**b**) Thermogram of array under normal condition at irradiance of 362 W/$m^{2}$.

**Table 1 sensors-23-06749-t001:** Characteristics of the PV system used in this study.

Parameter	Value
Array DC Power	3.36 kW
Latitude	10°32′
Longitude	84°31′
PV modules model	Canadian Solar CS6k-280M
Inclination	15°
Azimuth	150° with respect to the North

**Table 2 sensors-23-06749-t002:** Main characteristics of the thermal camera (FLIR VUE PRO R 336) used in the project.

Parameter	Value	Parameter	Value
HFOV × VFOV	25° × 19°	Image width × height	336 × 256
Sensor (widht × height)	5.764 mm × 4.351 mm	Accuracy	±5 °C
Focal Lenght	13.00 mm	Thermal sensitivity	40 mK

**Table 3 sensors-23-06749-t003:** Characteristics of the sensors used for measurements of environmental variables required for IRT analysis.

Component	Parameter	Value
DHT11 Sensor	Temperature range	0 °C to 50 °C
	Humidity Range	20% to 90%
	Accuracy	±2 °C and ±5%
Spektron 210	Irradiance range	0 to 1500 W/m${}^{2}$
	Accuracy	±5%

**Table 4 sensors-23-06749-t004:** Examples of tests applied to evaluate the performance of the system.

Figure	Irradiance (W/m${}^{2}$)	Module with Hot Spot Induced								Module with Hot Spot Detected							
		1	2	3	4	5	6	7	8	1	2	3	4	5	6	7	8
Figure 6a	317	1	0	1	1	0	1	0	1	1	0	1	1	0	1	0	1
Figure 6b	430	1	0	0	0	0	0	0	0	1	0	0	0	0	0	0	0
Figure 7a	456	0	0	0	0	1	0	0	0	0	1	0	0	1	0	0	0
Figure 7b	362	0	0	0	0	0	0	0	0	1	0	0	0	0	0	0	0

PV modules where the system detected the condition correctly are shown in green, and modules where an erroneous condition was detected are shown in red.

**Table 5 sensors-23-06749-t005:** Results of the confusion matrix of the experiment to test the system performance.

	True	False	∑
Positive	216	18	234
Negative	437	1	438
∑	653	19	672

## Data Availability

Data presented in this study are available on request from the corresponding author. The data are not publicly available due to internal policies of the Institution.

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
