# Peer review of "IoT System Based on Artificial Intelligence for Hot Spot Detection in Photovoltaic Modules for a Wide Range of Irradiances"

_sensors, 2023, doi:10.3390/s23156749_

Round 1

Reviewer 1 Report

The author proposes a photovoltaic module hot spot detection method and system. Locating the photovoltaic modules that may have problems is a very meaningful work, but the work presented by the author is relatively rough and lacks innovation, and major revision is recommended. The following are specific suggestions.

1. Authors should highlight the innovation of their work.

2. Why choose random forest instead of other algorithms to detect hot spots? Authors would do well to give comparative experiments.

3. Why not classify the factors that cause hot spots and then conduct training, and give the causes of hot spots while detecting hot spots?

4. The normal operation of photovoltaic is a more common state, and I think the false detection rate when working normally for a long time is also very noteworthy.

5. It is meaningless to simply detect hot spots, and the author should increase the discussion on the role and significance of detection hot spots for photovoltaic.

 Moderate editing of English language required

Author Response

  1. Authors should highlight the innovation of their work.

Section 2.4 is being introduced, where the innovation of the project is specifically highlighted.

  1. Why choose random forest instead of other algorithms to detect hot spots? Authors would do well to give comparative experiments.

Random Forest was chosen because it has been used in previous works, and the results obtained in our study were quite satisfactory. However, we believe that in the future, we could evaluate with more algorithms to achieve even higher accuracy. It is a good idea to consider evaluating different algorithms to obtain greater precision in our studies. While the Random Forest algorithm has yielded acceptable results in previous works, it is always beneficial to explore other options to ensure that we are using the most suitable approach for our problem.

  1. Why not classify the factors that cause hot spots and then conduct training, and give the causes of hot spots while detecting hot spots?

The observation you indicate is very valid, however, the main objective of this project is to demonstrate a new way of detecting hot spots from artificial intelligence for low irradiances, therefore, training the system to also identify the cause of the hot spot requires further development, which we are considering as future work. We are adding this aspect in the conclusions.

  1. The normal operation of photovoltaic is a more common state, and I think the false detection rate when working normally for a long time is also very noteworthy.

What you point out is very important. In our case, we have identified the causes of false alarms and will be working to improve this aspect. In this regard, we have made further information on this in section 4.4

  1. It is meaningless to simply detect hot spots, and the author should increase the discussion on the role and significance of detection hot spots for photovoltaic.

To expand on this aspect, section 1 was modified to include an additional bibliographic reference.

In the new version, the introduction, methodology, results and conclusions have been improved. New references were added and the wording and orthography of the entire document were improved.

Reviewer 2 Report

The paper presents an innovative Internet of Things (IoT) platform designed to automatically detect hot spots in photovoltaic (PV) modules through the analysis of temperature data. The authors have done a good job in presenting the research, as the content is well-organized and the methods and results are effectively communicated. The introduction provides a solid foundation for the study, effectively setting the context and highlighting the significance of the research topic. The references utilized in the paper are appropriately related to the research topic and appear to be up-to-date. However, there are several areas where the paper could be further enhanced:

1.      While the figures included in the paper are relevant and contribute to the understanding of the study, their resolution is low, making them difficult to read. To improve the visual experience for readers, it is recommended that the authors provide figures with higher resolution.

2.      The methodology section is well-presented, providing a clear explanation of the steps involved in detecting hot spots in PV modules using temperature analysis. The authors should consider including more details about any limitations encountered during the study.

3.      One aspect that remains unclear is the frequency at which the system implements measurements. It would be helpful if the authors could specify whether the measurements are taken hourly, daily, monthly, or at some other interval. Clarifying this information would provide a better understanding of the system's operational characteristics.

4.      The results presented in the paper are clear and provide valuable insights into the effectiveness of the proposed IoT platform. To further strengthen their findings, the authors could consider comparing their results with those obtained using one of the methods mentioned in Section 2. Such a comparative analysis would bolster the credibility of the proposed platform and highlight its advantages over existing approaches.

5.      The conclusion section effectively summarizes the main findings of the study and reiterates the significance of the research. It would be beneficial for the authors to include potential avenues for future research.

In summary, the paper is well-written and provides a valuable contribution to the field of PV module analysis using IoT technologies. By addressing the aforementioned suggestions, the authors can further improve the clarity, comprehensiveness, and impact of their work.

Author Response

  1. While the figures included in the paper are relevant and contribute to the understanding of the study, their resolution is low, making them difficult to read. To improve the visual experience for readers, it is recommended that the authors provide figures with higher resolution.

Improvements were made to the figures. Due to the specific resolution of the thermograms used, it is not possible to improve this. The sizes were increased to improve their visualization.

  1. The methodology section is well-presented, providing a clear explanation of the steps involved in detecting hot spots in PV modules using temperature analysis. The authors should consider including more details about any limitations encountered during the study.

Your observation is very important. Section 3.9 has been included where this information is detailed.

  1. One aspect that remains unclear is the frequency at which the system implements measurements. It would be helpful if the authors could specify whether the measurements are taken hourly, daily, monthly, or at some other interval. Clarifying this information would provide a better understanding of the system's operational characteristics.

To clarify this, section 3.4, and 4.4 has been updated.

  1. The results presented in the paper are clear and provide valuable insights into the effectiveness of the proposed IoT platform. To further strengthen their findings, the authors could consider comparing their results with those obtained using one of the methods mentioned in Section 2. Such a comparative analysis would bolster the credibility of the proposed platform and highlight its advantages over existing approaches.

This observation you make is very interesting. Because each investigation had particular considerations, it would not be possible for us at this time to make a rigorous comparison between the projects. We believe this could be the focus of future work.

  1. The conclusion section effectively summarizes the main findings of the study and reiterates the significance of the research. It would be beneficial for the authors to include potential avenues for future research.

The conclusions were updated to include future work to be carried out.

In summary, the paper is well-written and provides a valuable contribution to the field of PV module analysis using IoT technologies. By addressing the aforementioned suggestions, the authors can further improve the clarity, comprehensiveness, and impact of their work.

In the new version, the introduction, methodology, results and conclusions have been improved. New references were added and the wording and orthography of the entire document were improved.

Reviewer 3 Report

The authors used AI to detect hot spots of photovoltaic panels through Infrared thermography under irradiation between 300 - 700 W /m2.  The work can diagnose Photovoltaic (PV) installations.    The manuscript can not be accepted at the present status.

1. The authors should provide clarification on why their technique can detect hot spots at lower irradiation levels (down to 300 W/m2) compared to others that only work down to 700 W/m2.

2. The Introduction section needs revision. Specifically, consider moving lines 60-87 to the Materials and Methods section, and incorporating the "2. Related work" subsection into the Introduction. The Introduction is currently too lengthy.

3. Please mention the model of the IRT camera used.  In Table 2, some features such as Power Dissipation and Thermal Sensitivity may not be very important to the research and can be omitted to save space and readers' time.

4. Clarify the distinct functions of both the DS18B20 and DHT11 sensors for temperature measurement.  Also, explain how humidity affects the diagnostic process using AI.

5. The DHT11 sensor typically has a temperature measurement accuracy of ±2°C and a humidity measurement accuracy of ±5%. Please describe how the authors achieved an increased accuracy of ±1°C and ±1% in their research.

6. Provide clarity on the process of capturing the thermogram and transferring it to the Raspberry Pi (RPi).  Was all the infrared thermography (IRT) information analyzed on the RPi? Also, specify the model of the RPi used.

7. Explain how the authors integrated the results from deep learning techniques (ResNet50 and MobileNet) and machine learning (random forest).  It seems that using only one of these methods would be sufficient to identify hot spots.  Elaborate on the advantages and disadvantages of each method and highlight any differences in the results obtained from ResNet50, MobileNet, and random forest.

8. Briefly introduce the previous work referenced in Ref 10, including what has been accomplished and what remains to be explored.

Author Response

  1. The authors should provide clarification on why their technique can detect hot spots at lower irradiation levels (down to 300 W/m2) compared to others that only work down to 700 W/m2.

Your observation is very important to clarify. For this purpose, the introduction and section 3.5 were updated.

  1. The Introduction section needs revision. Specifically, consider moving lines 60-87 to the Materials and Methods section, and incorporating the "2. Related work" subsection into the Introduction. The Introduction is currently too lengthy.

In response to this observation, the introduction and methodology section was updated.

  1. Please mention the model of the IRT camera used.  In Table 2, some features such as Power Dissipation and Thermal Sensitivity may not be very important to the research and can be omitted to save space and readers' time.

To attend to this observation, table 2 and the text of section 3.2 were updated.

  1. Clarify the distinct functions of both the DS18B20 and DHT11 sensors for temperature measurement.  Also, explain how humidity affects the diagnostic process using AI.

To attend to this observation, section 3.2 were updated. Table 3 was outdated.

  1. The DHT11 sensor typically has a temperature measurement accuracy of ±2°C and a humidity measurement accuracy of ±5%. Please describe how the authors achieved an increased accuracy of ±1°C and ±1% in their research.

Thank you very much for your observation. Due to a typing error, the repeatability values were included instead of the accuracy. The table has been updated with the correct values.

  1. Provide clarity on the process of capturing the thermogram and transferring it to the Raspberry Pi (RPi).  Was all the infrared thermography (IRT) information analyzed on the RPi? Also, specify the model of the RPi used.

Raspberry 4.0 was used. IRT analysis is not performed on these. Section 3.4 was updated with this information.

  1. Explain how the authors integrated the results from deep learning techniques (ResNet50 and MobileNet) and machine learning (random forest).  It seems that using only one of these methods would be sufficient to identify hot spots.  Elaborate on the advantages and disadvantages of each method and highlight any differences in the results obtained from ResNet50, MobileNet, and random forest.

In response to this observation, we have made the changes, including the benefits of using the ResNet50 and MobileNet models, leveraging pre-trained learning to accelerate the classification of suboptimal conditions in photovoltaic panels. We have also highlighted the advantages of using random forest to identify points of interest in panels with optimal conditions. It's worth emphasizing that this is an initial exploration that has provided us with quite interesting results. In the future, we will validate a set of algorithms aimed at optimizing the good results obtained through this methodology. Your observation is very insightful and it helps us improve the classification process for the future.

  1. Briefly introduce the previous work referenced in Ref 10, including what has been accomplished and what remains to be explored.

In response to your comment, the introduction was updated to include more information on the previous stage of the project.

In the new version, the introduction, methodology, results and conclusions have been improved. New references were added and the wording and orthography of the entire document were improved.

Reviewer 4 Report

Title: A Review of IoT-Based Hot Spot Detection in Photovoltaic Modules Using Artificial Intelligence

Overall, the article "IoT-Based Hot Spot Detection in Photovoltaic Modules Using Artificial Intelligence" gives an in-depth look at how IoT and AI technologies may be used to identify hot spots in PV modules. The authors give experimental data and draw conclusions on the suggested system's usefulness. Here is my analysis of the paper:

Strengths:

1. Unique strategy: The research offers a unique strategy for detecting hot spots in PV modules by merging IoT and AI approaches. This integration has the potential to provide faster and more efficient detection than existing techniques.

2. Experimental Validation: The authors performed tests to validate the suggested system, yielding good findings with a sensitivity of 0.995 and an accuracy of 0.92. The inclusion of these experimental findings strengthens the research's credibility.

3. Practical applications: The report stresses the research's practical applications, stressing the possibility for solar installation managers and thermographers to detect hot areas at lower irradiances, allowing for more timely inspections and maintenance.

4. Potential Research Directions: The authors highlight numerous potential research directions, including increasing the training dataset and testing the system using thermograms acquired at various elevations and orientations. These proposals open the door to future investigation and enhancement of the suggested system.

5. Integration of IRT and AI: As a distinctive feature of this research, the report highlights the integration of infrared thermal (IRT) imaging and AI. The technique shows potential in recognizing hot patches generated by a variety of reasons, not only specific temperature patterns, by assessing temperature differentials at the pixel level.

Improvement Suggestions:

1. The methodology A more extensive discussion of the approach used would be beneficial to the study. Step-by-step descriptions of data collection, preprocessing, AI model training, and assessment would improve the research's repeatability and clarity.

2. Comparative Analysis: It would be beneficial to incorporate a comparison of the proposed system with existing methods for detecting hot spots in PV modules. This would offer a better understanding of the system's benefits and drawbacks in comparison to other systems.

3. Discussion of limits: While the authors accept the possibility of future enhancements, it would be beneficial to clarify the proposed system's limits more explicitly. Addressing possible problems, such as environmental fluctuations or hardware limits, would aid in gaining a more complete knowledge of the system's applicability.

4. Considerations for Real-World Deployment: Given the research's practical significance, detailing the feasibility and possible problems of deploying the suggested system in real-world circumstances would enrich the study. Cost, scalability, and interaction with current infrastructure should all be considered.

Overall, the study makes an important addition to the field of hot spot detection in PV modules by combining IoT and AI technology. The discovery, with its experimental validation and practical ramifications, offers up new paths for further investigation and application. The writers can improve the clarity and effectiveness of their work by addressing the proposed revisions.

Author Response

1. The methodology A more extensive discussion of the approach used would be beneficial to the study. Step-by-step descriptions of data collection, preprocessing, AI model training, and assessment would improve the research's repeatability and clarity

In response to this observation, the methodology section was updated.

2. Comparative Analysis: It would be beneficial to incorporate a comparison of the proposed system with existing methods for detecting hot spots in PV modules. This would offer a better understanding of the system's benefits and drawbacks in comparison to other systems.

In order to address the observation, section 2.4 was included

3. Discussion of limits: While the authors accept the possibility of future enhancements, it would be beneficial to clarify the proposed system's limits more explicitly. Addressing possible problems, such as environmental fluctuations or hardware limits, would aid in gaining a more complete knowledge of the system's applicability.

In order to address this comment, section 3.9 was included

4. Considerations for Real-World Deployment: Given the research's practical significance, detailing the feasibility and possible problems of deploying the suggested system in real-world circumstances would enrich the study. Cost, scalability, and interaction with current infrastructure should all be considered.

The conclusions section was updated to address this recommendation.

Overall, the study makes an important addition to the field of hot spot detection in PV modules by combining IoT and AI technology. The discovery, with its experimental validation and practical ramifications, offers up new paths for further investigation and application. The writers can improve the clarity and effectiveness of their work by addressing the proposed revisions.

In the new version, the introduction, methodology, results and conclusions have been improved. New references were added and the wording and orthography of the entire document were improved.

Round 2

Reviewer 1 Report

My concerns have been solved.

Reviewer 3 Report

It is a nice version.

Reviewer 4 Report

The authors have addressed several of the initial comments raised in the first round of review. The updates made to the introduction, methodology, results, and conclusions sections are commendable, and the inclusion of comparative analysis in section 2.4 adds valuable insights to the study.

Overall, the authors have made commendable progress in refining their work, and the improvements made to the manuscript demonstrate their commitment to addressing the initial comments raised. I am pleased to have this version proceed with publication. 

It is alright.